# The Potential Role of Regulated Cell Death in Dry Eye Diseases and Ocular Surface Dysfunction

**DOI:** 10.3390/ijms24010731

**Published:** 2023-01-01

**Authors:** Camilla Scarpellini, Alba Ramos Llorca, Caroline Lanthier, Greta Klejborowska, Koen Augustyns

**Affiliations:** Laboratory of Medicinal Chemistry, University of Antwerp, Universiteitsplein 1, B-2160 Antwerp, Belgium

**Keywords:** dry eye, regulated cell death, oxidative stress, ferroptosis, necroptosis, pyroptosis

## Abstract

The research on new treatments for dry eye diseases (DED) has exponentially grown over the past decades. The increased prevalence of dry eye conditions, particularly in the younger population, has received much attention. Therefore, it is of utmost importance to identify novel therapeutical targets. Regulated cell death (RCD) is an essential process to control the biological homeostasis of tissues and organisms. The identification of different mechanisms of RCD stimulated the research on their involvement in different human pathologies. Whereas apoptosis has been widely studied in DED and included in the DED vicious cycle, the role of RCD still needs to be completely elucidated. In this review, we will explore the potential roles of different types of RCD in DED and ocular surface dysfunction. Starting from the evidence of oxidative stress and inflammation in dry eye pathology, we will analyse the potential therapeutic applications of the following principal RCD mechanisms: ferroptosis, necroptosis, and pyroptosis.

## 1. Introduction

Considered one of the most prevalent eye syndromes, keratoconjunctivitis sicca, commonly known as dry eye disease (DED), affects millions of people worldwide, with a percentage ranging between 5 to 50% [1]. The Tear Film & Ocular Surface Society (TFOS) Dry Eye Workshop II (TFOS DEWS II) in 2017 defined dry eye as “a multifactorial disease of the ocular surface characterized by a loss of homeostasis of the tear film and accompanied by ocular symptoms, in which tear film instability and hyperosmolarity, ocular surface inflammation and damage, and neurosensory abnormalities play etiological roles” [2]. In the past years, the impact of this disorder has increased worldwide, particularly among the younger population [3]. The major risk factors are frequent use of video screens, environmental causes such as pollution and low humidity, and wearing of contact lenses [4]. The symptoms can be mild or more severe, generally leading to discomfort and visual disturbance [5]. Epidemiological studies underline that age and sex have an impact on the symptoms and signs of DED [6]. Indeed, hormones seem to influence the incidence of eye disorders in the population, particularly DED [7]. The female sex is considered a risk factor with a prevalence between 10–20% of DED, especially in the post-menopause stage [8]. During menopause, reduced estrogen levels promote evaporative DED [9]. However, recent studies hypothesised that menopause evaporative DED was actually due to the reduction of androgens levels rather than estrogens [7,10,11,12]. Therefore, additional studies are needed to further elucidate the role of estrogens in DED [13]. In addition, iatrogenic intervention is one of the most studied causes of DED [14]. In iatrogenic DED, the common risk factors are the use of topical drugs (mainly linked to the presence of preservatives) [15], systemic medications (DED can be a consequence of adverse effects) [16], contact lenses wear, and ophthalmological surgical procedures (related to the procedure itself or as a transitory/permanent side-effect) [14]. 

Based on an etiological classification, DED can be subdivided into Aqueous Deficient Dry Eye (ADDE) and Evaporative Dry Eye (EDE) [17]. In ADDE, lacrimal tear secretion and volume are reduced, leading to hyperosmolarity and tear film instability [18]. Consequently, the inflammatory mediators are released and detected in the tears of patients suffering from ADDE, particularly in patients affected by Sjögren syndrome dry eye (SSDE) [19]. The Sjögren syndrome is a multifactorial autoimmune endocrinopathy associated with immunologic abnormalities, and is characterised by a severe form of dry eye and/or dry mouth [20]. In EDE, normal lacrimal secretion is followed by excessive evaporation of the tear film, leading to tear hyperosmolarity. The most important aspect of EDE is the meibomian gland dysfunction (MGD), which alters lipid secretion perturbating the corresponding lipid layer of the ocular surface [21]. Consequently, the tear components are modified and the aqueous layers evaporate rapidly [22]. The alteration in the electrolytes equilibrium can cause damage to the ocular surface and tear film, promoting hyperosmolarity and oxidative stress [23]. As a consequence, the concentration of reactive oxygen species (ROS) on the ocular surface increases, propagating dry eyes into a vicious circle [24,25,26]. The tear hyperosmolarity is the main factor responsible for the activation of different pathways leading to the release of inflammatory mediators and proteases [24,27].

Dry eye disease concerns ocular surface composed of cornea, conjunctiva, tear film, lacrimal glands, meibomian glands, eyelids, muscles, and nerves (Figure 1). A healthy human tear film is composed of lipids, water, mucin, proteins, electrolytes, and vitamins [28]. The tear-secreting glands, the lacrimal and meibomian glands, along with the goblet cells in the eye and eyelid, produce the tears and work with the ocular surface to maintain optimum ocular health by lubricating the eye, removing debris, and protecting from infection [29]. The transparent, dome-shaped cornea presents superficial microvilli which aid the tear film anchorage and regulates the secretion of growth factors and cytokines as a shield [30]. The conjunctiva, situated between the corneal rim and the lid margin, is composed of goblet cells responsible for mucin secretion [31]. Externally, the eye is coated with the tear film, formed by three different regions: an external lipid layer, a mild aqueous layer, and an inner mucin layer [32]. The lipid layer is regulated by the meibomian glands lipids and proteins secretion and the main role is to prevent evaporation of the aqueous layer [33]. The lacrimal glands regulate the water, electrolytes, proteins, and mucus content in the aqueous phase, which is extremely important for eye irritation [34]. The electrolytes play a significant role to prevent hyperosmolarity and alter physiological conditions [35]. Additionally, the eyelid contributes to prevent the desiccation of the ocular surface [36]. Internally there is a mucin layer which acts as a surfactant spreading homogeneously through the tear film of the ocular surface [37]. Secreted by the conjunctival goblet cells, the mucus is mainly based on enzymes, mucins, and leukocytes. They also minimise friction and protect the cornea during blinking [36].

The diagnostic methods currently used in DED diagnosis are based on a questionnaire called Dry Eye Questionnaire-5 (DEQ-5) or Ocular Surface Diseases Index (OSDI), collecting all the information to indicate the positivity of DED [38]. Patients normally report burning, photophobia, itching, and foreign body sensation [39,40]. However, the quantification of the symptoms is based on the tear film instability measurement (break up time (BUT)), the tear osmolarity assessment, and the tear volume measurement (Schirmer’s test) [41]. The next step is the discrimination between the two subtypes, ADDE and EDE, followed by medical prescription of the treatment according to the severity of the symptoms [42]. 

One of the therapeutic strategies is the application of artificial tear substitutes to improve lubrication and decrease evaporation to provide temporary relief. Different formulations, such as eye drops, topical lubricants, gels, and ointments, are available on the market [43]. The first medication used to treat DED is Cyclosporine A ophthalmic emulsion 0.05% (*Restasis*^®^, Allergan, Irvine, CA, United States) [44]. To ameliorate the drug delivery into the eye, the novel nanomicellar formulation was used for Cyclosporine A ophthalmic solution 0.09% (*Cequa*^®^, Sun Pharmaceutical Industries, Cranbury, NJ, United States) [45]. Cyclosporine acts as an immunomodulator for severe DED, reducing inflammation markers and cell death on the ocular surface [46]. Recently, Lifitegrast ophthalmic solution 5% (*Xiidra*^®^, Novartis Pharmaceuticals Corporation, Basel, Switzerland) was the first medication approved in the United States for the treatment of DED signs and symptoms [47]. Lifitegrast A inhibits the release of cytokines, interferon δ, TNF-α, and other interleukins preventing the activation of the ocular inflammatory cycle [48]. An additional corticosteroid drug is Loteprednol etabonate ophthalmic suspension (*Eysuvis*^®^, Kala Pharmaceuticals, Wtaertown, MA, United States) approved for the treatment of inflammatory flares [48]. Additionally, in 2021, the Food and Drug Administration (FDA) approved Varenicline (*Tyrvaya*^®^, Oyster Point Pharma, Princeton, NJ, United States), a nasal spray which stimulates tears, mucins, and oil production and treat both, signs and symptoms [49]. A novel treatment in phase III of clinical trials, NOV03 is a preservative-free eye drop formulation that alleviates the dryness of the ocular surface [50]. The necessity for novel preservative-free formulations has grown in the last years due to the DED increased risk linked with benzalkonium chloride (BAK), a well-known quaternary ammonium compound commonly used in the formulations of numerous ophthalmic preparations [26,51,52]. However, the current, research is focused on the development of novel devices, processes, and medicines to promote tear secretion rather than substitute them with medications or eye drops [53]. Therefore, the identification of additional biochemical pathways involved in the pathophysiology of DED is fundamental to specifically address the research of novel treatments. 

## 2. Oxidative Stress and Inflammation in DED

Oxidative stress is considered one of the main hallmarks of dry eye, leading to inflammation and general discomfort [26,54,55,56]. An imbalance between ROS production and the antioxidant capacity of the organism results in oxidative stress. Consequently, ROS accumulation can be responsible for damage at different levels and particularly on the membrane and ocular surface [57,58,59]. The reactive hydroxyl radical can initiate the lipid peroxidation of cell membranes and induce the accumulation of lipid hydroperoxides leading to membrane disruption and ultimately cell death [60]. Increased levels of lipid hydroperoxide are detected by two major biomarkers, malondialdehyde (MDA) and 4-hydroxynonenal (4-HNE) [20]. The presence of both MDA and 4-HNE have been detected in the tear film and ocular surface in DED patients [61,62]. The increased concentration of lipid hydroperoxides seems to correlate positively with other parameters detected in DED diagnosis, such as tear film break-up time and Schirmer tear volume [25].

Following ROS -induced lipid peroxidation and in response to cellular stresses, mitogen-activated protein (MAP) kinases and nuclear factor kappa- B (NF-kB) are activated and mediate the release of inflammatory cytokines IL-1α, IL-1β, tumour necrosis factor-alpha (TNFα) and metalloproteinase 9 (MMP-9) [63,64,65]. Moreover, the inflammatory response in cornea and conjunctival cells is enhanced by T-cell upregulation of CD3+, CD4+, and CD8+, and the release of additional pro-inflammatory cytokines such as IL-6 and IL-8 [66,67]. Although the implication of inflammation in DED has been known for more than 40 years, its role as cause and effect of DED was established only recently [68]. Moreover, inflammation is also included in the new concept of the ‘‘vicious cycle of inflammation” (Figure 2) proposed by Baudouin et al., where tear film instability, tear hyperosmolarity, apoptosis of corneal/conjunctival cells, and inflammation in the ocular surface are the key features [24]. Breaking the cycle by targeting one of the main factors is considered a valid therapeutic strategy [24]. Starting with tear film instability and hyperosmolarity, the quality of tears can be affected by a decrease of the water content or an increase of evaporation [69]. As a consequence, inflammatory mediators are released in the tear fluid resulting in the release of inflammatory cytokines, leading to damage to the corneal and conjunctival cells and consequent cell death through apoptosis [70]. All these features aggravate the inflammation and maintain the vicious cycle promoting hyperosmolarity and tear film instability.

Based on the implication of oxidative stress and inflammation in dry eye etiopathology and considering the role played by apoptosis in the dry eye vicious cycle, the possible involvement of different types of regulated cell death (RCD) in DED and other ocular surface diseases was investigated [71,72]. In the first part of this review, we introduce the concept of RCD starting from apoptosis. In the second part, the possible role of the most investigated types of RCD, ferroptosis, necroptosis, and pyroptosis, is examined, as well as their potential as a therapeutic target for the treatment of DED. The specific biochemical mechanisms and the hallmarks for ferroptosis, necroptosis, and pyroptosis are described together with DED ocular surface biomarkers (Table 1).

## 3. Regulated Cell Death (RCD): An Emerging Field

Cell death is considered one of the most important mechanisms to maintain cell homeostasis and integrity of multicellular organisms [78]. Primarily observed by Carl Vogt in 1842, it has been studied for more than two centuries [79]. Over the past decades, many experimental evidence unveiled and characterised a detailed set of endogenous genes involved in the cell death machinery [80]. In 1964, Lockshin and Williams introduced the concept of “programmed cell death” (PCD), based on the observed cells capacity to induce a genetically regulated self-destruction [81,82,83]. In 1972, with the definition of the term “apoptosis” by John Kerr and his group, the concept of “regulated cell death” emerged [84]. Traditionally, a distinction was made between regulated (RCD) and accidental cell death (ACD), based on the different morphological features [80]. Specifically, RCD was used as a synonym for apoptosis, while ACD referred to necrosis [85]. While RCD can undergo specific morphological modification that can be potentially targeted, ACD occurs in response to extreme physical stresses with a nonspecific response preventing any molecular intervention [86]. Since 2005, the Nomenclature Committee on Cell Death (NCCD) formulated guidelines for the classification of different cell death types [87]. They proposed unified criteria that moved from the historical oversimplistic classification of RCD and ACD to a more updated classification based on events associated with specific cell death modalities, their biological context, and effectors [80]. With an increased number of research groups working in the field of regulated cell death, the importance of this phenomenon is now widely recognised and many novel types of non-apoptotic RCD have been identified over the last 20 years [85,88].

Among RCD, apoptosis and inflammatory RCD can be separately distinguished [89]. Apoptosis is characterised by cell shrinkage, with consequent reduction of cellular volume, chromatin condensation, membrane blebbing, nuclear fragmentation, and the final separation of the cellular components into apoptotic bodies collected by neighbouring cells with phagocytic activity and degraded by lysosomes [80,84]. These specific biochemical features distinguish it from the other RCD. More importantly, in apoptosis, there is no disruption of the cellular membrane and, therefore, no inflammation process occurs, whereas inflammatory RCD is characterised by an inflammatory response induced by cell membrane lysis and release of cytosolic material [90]. The most widely studied RCDs are: ferroptosis, an iron-dependent form of regulated cell death driven by enhanced lipid peroxidation which reduces the organism antioxidant defence-inducing cell damage [91]; necroptosis, a programmed form of necrosis dependent on RIPK1-RIPK3 activation of MLKL [92]; pyroptosis, an inflammasome gasdermin D dependent form of RCD [93]. 

The role of apoptosis in DED has been investigated for several years. A correlation between ocular surface inflammation and the expression of proapoptotic markers (Fas, Fas ligand, APO2.7, CD40, and CD40 ligand) has been found [94]. In chronic dry eye in human and dog models, apoptosis has been demonstrated [70]. Recently, the upregulation of apoptosis in human corneal epithelial cells (HCEpiCs) because of hyperosmolarity has been verified [95]. Vitoux M.A. et al. demonstrated that BAK can induce apoptosis in a concentration-dependent manner in HCE [96]. Whereas, a recent study reported BAK not only as oxidative stress and apoptosis inducers, but also as a trigger of emerging cell death mechanisms [97]. 

More importantly, apoptosis is included in the vicious cycle of DED as reported by Baudouin and co-workers, underlying the relevant role of conjunctival and corneal cell death in the etiopathology of dry eye [98]. Therefore, the possible implication of the inflammatory RCDs ferroptosis, necroptosis, and pyroptosis as a potential therapeutic target for DED is also examined.

## 4. Targeting RCD as an Innovative Strategy in DED

### 4.1. Ferroptosis

Ferroptosis is an iron-catalysed non-apoptotic form of cell death initially described in RAS-mutated oncogenic cells and more recently connected with the pathophysiological processes of many diseases in different vital organs (Figure 3) [99,100]. The key features of ferroptosis are the accumulation of lipid peroxides, abnormal iron metabolism, and reduced levels of glutathione (GSH) as well as GPX4, which can lead to a reduced antioxidant defence of the cells, damage on the phospholipids bilayers, and consequently, cell death [101,102]. Specifically, during lipid peroxidation, a chain reaction of bis-allylic hydrogen abstraction and oxygenation of polyunsaturated fatty acids (PUFAs) of phospholipids, catalysed by redox-active iron, results in the accumulation of toxic lipid hydroperoxides [103]. Iron availability is regulated by two main sources: ferrous iron (Fe^2+)^ in the cytosolic labile iron pool (LIP) and in the catalytic centres of non-heme iron proteins, e.g., lipoxygenase (LOXs) [104]. The former leads to non-enzymatic random oxidation based on Fenton reaction, whereas the latter gives enzymatic oxidation of specific substrates [105]. In addition, the enzyme GPX4 can control iron-dependent lipid peroxidation by the reduction of reactive lipid peroxides in their corresponding inactive lipid alcohols [101]. In order to exert its mechanism of action, the selenocysteine GPX4 requires two electrons provided by the intracellular cofactor GSH, synthesised from cysteine [106]. Cysteine availability is regulated by cystine/glutamate antiporter (System Xc-), which transports cystine inside the cytosol in exchange for glutamate in a 1:1 ratio [107]. Activating or blocking the ferroptosis pathway to alleviate the progression of the disease provides a promising therapeutic strategy [100]. Recent studies reported the implication of ferroptosis in the occurrence and progress of many diseases, such as tumours, neurological diseases, acute kidney injury, and ischemia/reperfusion [108,109,110,111,112]. There are different eye disorders where the implication of lipid hydroperoxide and GPX4 have been underlined, without proving the ferroptosis role. Whereas the presence of the main hallmarks has been detected in retinal cells, photoreceptors, cornea, and conjunctival cells, ferroptosis has been confirmed in the ethiopathology of only a few conditions [113]. Therefore, further research should be undertaken to investigate the potential central role of ferroptosis in other eye disorders [114].

#### Possible Implication of Ferroptosis in Dry Eye and Ocular Surface Dysfunction

ROS accumulation and modification of the lipid layer on the ocular surface are key features in the pathogenesis of DED as described in Section 2. Oxidative stress and inflammation in DED [115]. Different studies underlined how the loss of cell functions is an important contribution to the alteration of the ocular barrier [35,116,117]. We hypothesise, therefore, that ferroptosis can be considered as a key mechanism in the etiopathology of DED.

The cornea is exposed to many different external and environmental factors [118]. Ultraviolet radiation (UV) and global warming are among the causes that can promote oxidative stress and, consequently, cornea cell dysfunction [114,119]. GPX4 is one of the enzymes that maintains redox homeostasis and promotes wound healing [115,120]. As described in Section 4.1 Ferroptosis, reduced availability of GSH or GPX4 promotes lipid peroxidation and decreases regeneration of the corneal epithelium [121]. The addition of α-tocopherol, a lipophilic antioxidant, and known ferroptosis inhibitor, significantly improved the delay in wound healing [120]. This study suggested how ferroptosis inhibition can help to protect the corneal cells. Recently, Katsinas N. et al. demonstrated the antioxidant and anti-inflammatory activity of a phenolic extract from olive pomace, which can control the imbalanced ROS formation [122]. In an additional study reported by Lovatt M. et al., the use of another well-known ferroptosis inhibitor, Ferrostatin-1, in the Fuch’s endothelial corneal dystrophy (FECD) could prevent the accumulation of lipid peroxides [123]. Recently, Yuan J. and co-worker reported a study where excessive ROS production induced ferroptosis in a DED model [124]. They verified the upregulation of the enzyme aldo-keto reductases (AKR1C1), which is responsible for detoxifying 4-HNE in its corresponding non-toxic byproduct. Moreover, they detected a decrease of the common inflammatory biomarkers expression (TNFα, IL-1β) after Ferrostatin 1 treatment in a mouse model. However, partial ferroptosis was detected in HCE cells under hypertonic conditions and AKR1C1 overexpression. Certainly, additional studies are required to fully confirm the potential role of AKR1C1 in ferroptosis and DED. 

The tear fluid contains several antioxidants able to protect the ocular surface from different diseases connected with ROS accumulation [125]. In patients suffering from Sjogren’s syndrome, where there is a lack of tear fluid, a high level of late and early-stage lipid peroxidation biomarkers such as 4-HNE and hexanoyl-lysine (HEL) have been detected [20]. ROS overexpression might be induced by exposure to atmospheric oxygen and/or alteration of antioxidant support, contributing to tear film instability and consequent ocular surface damage and inflammation [58]. Among the most common damage caused by ROS, lipid peroxidation of the membrane, oxidative stress of protein, and oxidative damage of DNA have been highlighted [126]. Wakamatsu T.H. et al. detected an increased lipid peroxidation in the tear film and conjunctival cells [59]. Unfortunately, very little is described regarding the role of epithelial lipids in the cell death process and inflammation involved in eye disorders, particularly in DED. In a recent study reported by Magny R et al., lipid markers were analysed as a consequence of BAK and hyperosmolarity inflammation [127]. In patients suffering from dry eye, an alteration of the cornea epithelial cells due to the accumulation of ROS was observed, reducing the differentiation capacity of the cells and hindering the blinking mechanism [128]. Possibly, the use of a radical scavenging agent might be an interesting therapeutic approach, particularly lipid radical trapping antioxidants (RTAs) could tackle the accumulation of lipid peroxides [129]. 

The exposure of the ocular surface to oxidative stress together with UV light and environmental stress is also responsible for many dysfunctions of the conjunctiva [118,130]. Dry eye, atopic keratoconjunctivitis, and conjunctivochalasis are only a few of the pathologies that can affect the ocular surface, caused by an alteration of the redox balance [31,58,127,131,132]. Together with other antioxidative enzymes such as GPX1, and superoxide dismutase (SOD1 and SOD2), GPX4 controls redox homeostasis [121,133,134]. Specifically, a reduction of its expression could lead to lipid ROS accumulation, promoting cell deaths in conjunctival cells and diseases associated with oxidative stress. Although apoptosis is the main mechanism of cell death associated with GPX4 loss, the recently discovered ferroptosis could also be considered as a possible mechanism involved in the alteration of the conjunctival cell and different eye diseases [135]. Sakai O. et al., verified the fundamental role of GPX4 in human conjunctival epithelial cells to maintain oxidative homeostasis and protect conjunctival cells from cytotoxicity [136]. Although further evidence is needed, all the elements above suggest that ferroptosis and GPX4 could be novel therapeutic targets for dry eye disease and other ocular surface diseases.

The correlation between ferroptosis and other types of eye diseases is beyond the scope of this review. However, it is relevant to mention that previous studies detected the presence of ferroptosis hallmarks in age-related macular degeneration (AMD) [137,138,139,140,141,142], glaucoma [113,143], retinitis pigmentosa [113,144,145], cataract [132,146], Retinal Ischemia-Reperfusion Injury (RIRI) [113,147], and alkali burn [148].

### 4.2. Necroptosis

Necroptosis is a caspase-independent form of regulated cell death that can be triggered by TNFα, toll-like receptors TLR3-TLR4, interferon receptor 1 (IFNAR1), Z-DNA binding protein 1 (ZBP1), and Fas (Figure 4) [149]. Necroptosis is regulated by the activity of receptor-interacting serine/threonine kinase 1 (RIPK1), a serine/threonine protein kinase which can be activated through the phosphorylation of its kinase domain. Activated RIPK1 then recruits RIPK3 through the interaction with the conservative RIP homotypic interaction motif (RHIM) which forms the so-called Necrosome [150]. Then RIPK3 phosphorylates the Thr357 and Ser358 residues of downstream protein mixed lineage kinase domain-like (MLKL), which oligomerises and translocates to the plasma membrane [151]. Here, pMLKL can execute necroptosis, inducing direct rupture of the plasma membrane, and form a pore-complex to promote leakage of intracellular material, or can dysregulate the Na^+^ and Ca^2+^ channels, leading to increased intracellular osmotic pressure and final cell death [152,153]. The membrane permeabilization induces the release of different pro-inflammatory cytokines, promoting inflammation [92]. Moreover, mitochondrial ROS production contributes to necroptosis induction, although the exact mechanism is not completely understood [154]. RIPK1 can be therapeutically targeted in order to inhibit necroptosis and the consequent inflammatory process. In different organs and pathologies, necroptosis plays a fundamental role and, therefore, has been widely investigated since 2005 [155]. Upregulation of RIPK1 was detected in ischaemia-reperfusion injury (IRI), atherosclerosis, acute kidney injury, and neurodegenerative diseases [156,157,158,159,160,161]. Recently, different studies revealed the possible implication of necroptosis in ocular conditions, with particular attention paid to corneal surface dysregulation [144,162,163,164]. Additionally, RIPK1 also regulates cell survival and a form of RIPK1-dependent apoptosis (Figure 4) [165].

#### Possible Implication in Dry Eye and Ocular Surface Dysfunction

Recently, Shi K. et al. reported the implication of necroptosis in airborne particulate matter (PM) ocular surface injury [71]. Necroptosis can induce cell death in the cornea epithelial cells following PM exposure. Currently, this is the only study that verified necroptosis on the ocular surface, demonstrating a potential link between necroptosis and DED. Indeed, DED is the most frequent clinical condition induced by PM exposure [71]. Treatment with Nec-1 successfully increased cell viability and reduced ROS formation, confirming the potential of using necroptosis inhibitors to prevent cell death. Moreover, considering the etiopathology of DED, it appears that inflammation is one of the main features together with the redox imbalance [63]. Moreover, TNF-α is a well-described mediator of DED pathophysiology and it is also indicated as an initiator of necroptosis. Particularly, a dramatic increase of ROS level is observed when TNF-α interacts with its receptor TNFR1 [166]. Additionally, in a recent study reported by Kessal K. et al., upregulation of RIPK1 was detected in a mouse dry eye model. However, the role of necroptosis in the inflammatory cascade was not confirmed [167]. Whereas the release of pro-inflammatory cytokines such as IL-1α, IL-1β, IL-6, and IL-8 and consequent activation of MAPK kinases is verified, there are no studies available regarding the involvement of RIPK1-RIPK3-MLKL and necroptosis [168].

Necroptosis is also implicated in different eye diseases. Particularly, the use of RIPK1 inhibitor Necrostatin 1 emerged as a promising therapeutic strategy to control the necroptosis-mediated inflammatory process in dry AMD [113,164,169,170], glaucoma [162,163], and retinitis pigmentosa [144,171]. 

### 4.3. Pyroptosis

The term pyroptosis, from the Greek words “pyro” (fire or fever) and “ptosis” (falling), described a novel type of RCD discovered in 2001 by B. Cookson and co-workers (Figure 5) [88,172,173]. The main feature of pyroptosis is the pore formation mediated by gasdermin D (GSDMD) and the activation of the inflammatory response in a caspase 1-dependent (canonical) or independent (non-canonical) way. After stimulation, cells form a cytosolic multiprotein complex, called inflammasome, among which, NOD-like receptor pyrin 3 (NLRP3) inflammasome is the most investigated [174]. NLRP3 is responsible for the release of inflammatory interleukins (IL-1β and IL-18), the formation of an apoptosis-associated speck-like protein (ASC), and the activation of pro-caspase 1. Particularly, caspase 1 (CASP-1) mediates the maturation of pro-IL-1β and pro-IL-18 in their corresponding mature form IL-1β and IL-18, and the cleavage of GSDMD [175]. In the canonical pyroptosis pathway, pathogens can trigger the NLRP3 inflammasome aggregation, lead to CASP-1 activation and GSDMD cleavage [176,177]. The CASP-1-dependent inflammasomes are divided between NLR and non-NLR inflammasomes which can be activated selectively by pathogen-associated molecular patterns (PAMPs) or damage-associated molecular patterns (DAMPs) [178]. Whereas, in the non-canonical pyroptosis pathway, a priming signal represented by microbial lipopolysaccharide (LPS) recognised by TLR4 or endogenous molecules like TNF-α stimulate the oligomerisation of caspase 4 or 5 (in human) or caspase 11 (in mouse) and then cleavage of GSDMD [179]. For both canonical and non-canonical pyroptosis, the final step is the cleavage of GSDMD in the N-terminal fragment (GSDMD-N), which can generate pores on the phospholipids of the plasma membrane and lead to the consequent cell death [180]. In addition, GSDMD cleavage from the non-canonical pathway can also promote an amplification of pyroptosis by stimulating the inflammasome release of CASP-1, the release of IL-1β and IL-18 and activates the inflammatory response [181]. Particularly the latter, can recruit interferon-gamma (IFN-γ) and, therefore, enhance the immune response [182]. In addition, caspase-8 activation by RIPK1 was also recently reported to induce cleavage and activation of the pore-forming molecule GSDMD in macrophages, thereby promoting the induction of a specific form of pyroptosis [183].

#### Possible Implication in Dry Eye and Ocular Surface Dysfunction

Different cornea disorders can be connected with pyroptosis-pathway activation [162]. In keratitis, which can cause cornea ulcers and consequent blindness, inflammasomes are formed as exogenous infections mediated by DAMPs signalling [184]. Mouse models infected with streptococcus pneumonia and pseudomonas aeruginosa can trigger NLRP3 or NLRC4 inflammatory response [185]. Consequently, caspase 2 can activate the inflammatory mediators leading to the cleavage of GSDMD. Additionally, the non-canonical pyroptosis pathway is involved in the p. aeruginosa keratitis [186]. Targeting caspase 4/5/11 could be a strategy to control the inflammatory response and prevent GSDMD cleavage [187]. In alkali burn, which can affect permanently the cornea transparency, activation of NLPR3 inflammasome and IL-1β were detected and successfully reduced by the use of NLRP3 inhibitors, underlying the role played by pyroptosis in this condition [188]. 

The elevated level of GSDMD-N detected in dry eye patients supports the hypothesis of pyroptosis implication in DED [189]. Environmental factors, and particularly fine particulate, can induce corneal pyroptosis and, therefore, promote dry eye [190]. A desiccating stress-induced dry eye mouse model with elevated levels of NLRP3 inflammasome, ASC, and CASP1 has been detected together with mature IL-1β and IL-18 [191]. In addition, GSDMD-driven pyroptosis mediated by TLR4 activation has been demonstrated in a mouse model [192]. In vitro, the increased concentration of ROS and hyperosmotic stress promote pyroptosis-mediated inflammation, which was tackled by disulfiram and calcitriol, two pyroptosis inhibitors [193]. The dry eye signs, such as tear loss or inflammatory response, can be aggravated by NLRP12/NLRC4 inflammasome activation mediated GSDMD cleavage together with IL-33 and IL-1β [191]. In dry eye patient tears, the administration of calcitriol effectively alleviates the hyperosmotic stress induced by NLRP3-ASC-CASP1-GSDMD pyroptosis cascade [193]. The administration of dexamethasone attenuated pyroptosis in DED in vitro and in vivo models with decreased expression of inflammatory factors. However, the specific mechanism of action still needs to be completely elucidated [194]. The expression of inflammasome AIM2 and elevated levels of IL-1β was detected in patients suffering from Sjögren’s syndrome [195].

The role of pyroptosis in various eye diseases is also widely studied. The presence of the classical hallmarks are detected not only in vitro, but also in an in vivo rat model, particularly for AMD [196,197,198,199,200]. Additionally, in glaucoma [200,201], cataract [202], and uveitis [203], the implication of pyroptosis was determined. However, different from ferroptosis and necroptosis, the implication of pyroptosis in retinitis pigmentosa is not clarified yet [204].

## 5. Conclusions

With a higher incidence of dry eye in the worldwide population, the necessity of discovering novel targets and biological pathways involved in the etiopathology of DED has emerged. The vicious cycle is a well-established concept that includes the main features of DED: tear film instability, tear hyperosmolarity, apoptosis, and inflammation. Breaking the vicious cycle is one of the most investigated strategies that can help to alleviate dry eye symptoms and the development of novel therapies. Based on the anatomy of the ocular surface, corneal and conjunctival epithelia are the most exposed to external factors, which can lead to a reduction in the thickness of the external lipid layer and ROS accumulation followed by inflammation and cell death. Among the different cell death mechanisms, only apoptosis has been included in the etiopathology of DED and the vicious cycle [1]. However, considering the complexity of DED, it is reasonable to investigate additional cell death mechanisms. Particularly, the roles of ferroptosis, necroptosis, and pyroptosis are emerging in different ocular diseases and have been verified directly to DED in the case of necroptosis and pyroptosis. The hypothesis of a crosstalk among ferroptosis, necroptosis, and pyroptosis cannot be excluded and left aside, considering the overlay for most of the biochemical pathways involved. Moreover, different studies have underlined the effectiveness of targeting regulated necrosis in various ocular diseases, and more recently, in cornea disorders. Taking all these considerations together and keeping in mind the crosstalk between the redox imbalance and inflammation in DED, as well as the link with the different RCD mechanisms, the research on regulated cell death in DED might be a novel area of study to identify novel therapeutic targets and develop novel therapies for patients.

## Figures and Tables

**Figure 1 ijms-24-00731-f001:**
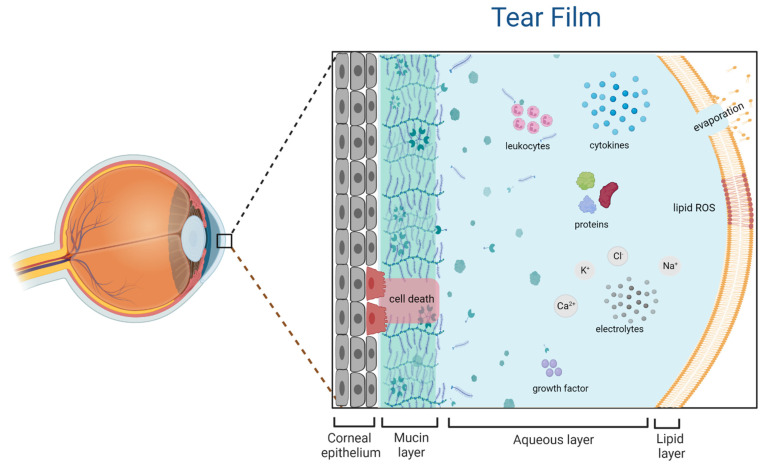
Representation of the ocular surface and tear film composition. The tear film is constituted by the external lipid layer which prevents evaporation in physiological conditions, the aqueous layer, responsible for lubrication, and the nutrients content to maintain osmolarity, and the mucin layer which protects the internal corneal epithelium.

**Figure 2 ijms-24-00731-f002:**
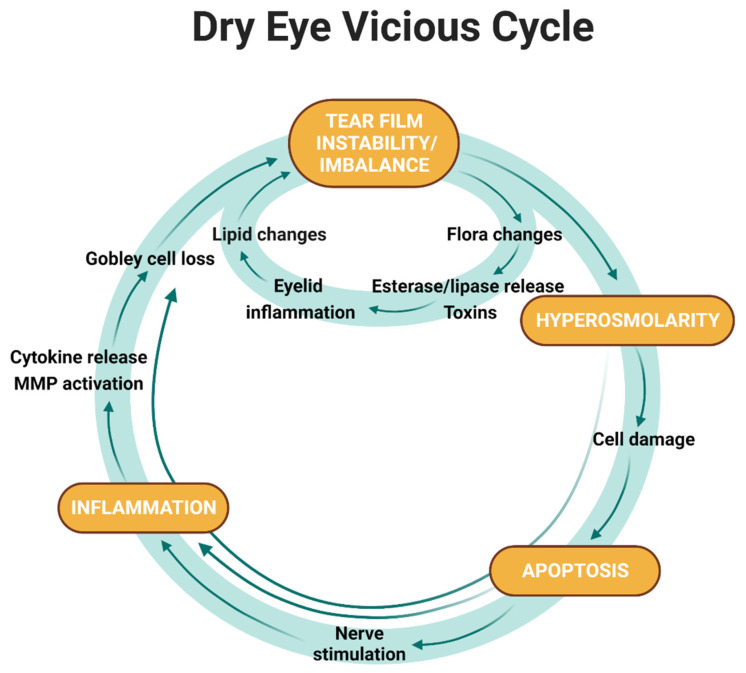
Representation of the vicious cycle of inflammation. Figure adapted from Baudouin C. et al. [24].

**Figure 3 ijms-24-00731-f003:**
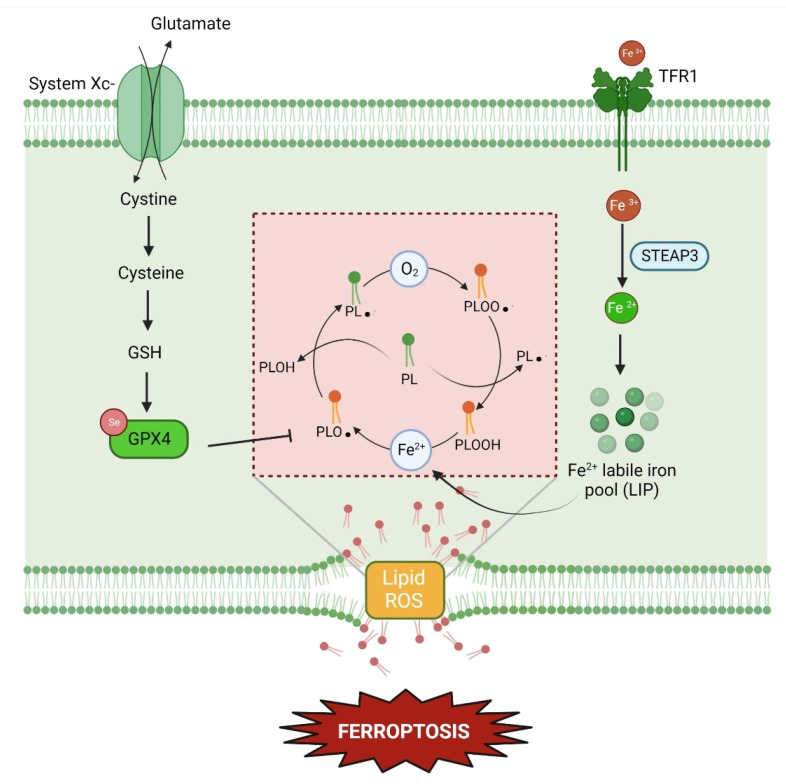
Overview of ferroptosis pathway. The system Xc- exchanges glutamate and cystine in a 1:1 ratio. From cystine, GSH is synthesised and available as a substrate for GPX4, responsible for controlling the lipid peroxidation process. Transferrin is responsible for the intake of Fe^3+^, which is transformed into its reduced form ferrous iron (Fe^2+^) by the metalloreductase STEAP3. Fe^2+^ constitutes the labile iron pool and mediates lipid peroxidation via the Fenton reaction. The lipid peroxides chain process is depicted in the central area of the figures. PL = phospholipids; PL• = phospholipid radicals; PLOOH = phospholipid hydroperoxides; PLO• = alkoxyl phospholipid radicals; PLOO• = phospholipid peroxyl radicals; PLOH = phospholipids alcohols.

**Figure 4 ijms-24-00731-f004:**
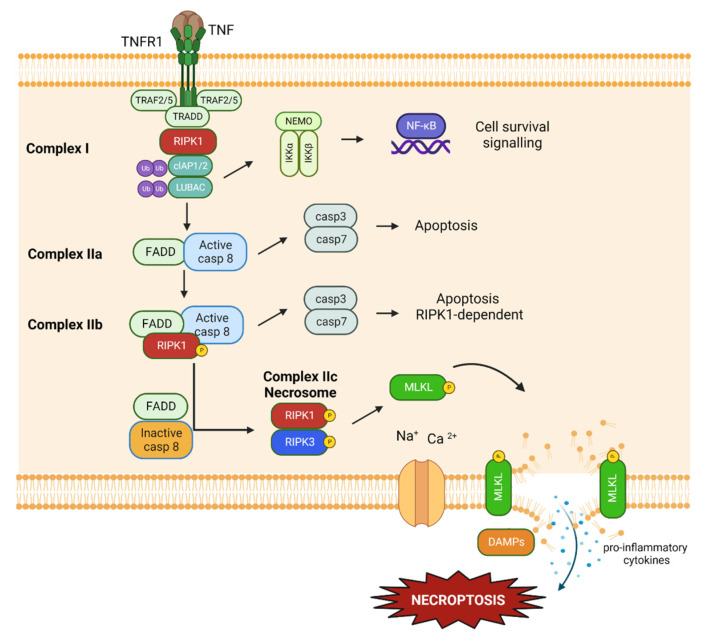
Overview of the TNF-mediated RIPK1 activation pathway. (1) RIPK1 can mediate cell survival when cIAP1/2 and LUBAC polyubiquitinate different components of TNFR1-complex I, resulting in the downstream activation of TAB2/TAB3/TAK1 and NEMO/IKKα/IKKβ complex. The activation of IKKα/β can promote cell survival by NF-kB-dependent upregulation of pro-survival genes. (2) When complex IIa is formed, it can mediate apoptosis. CYLD deubiquitinates RIPK1 which subsequently is released from complex I. The TRADD-RIPK1 complex recruits FADD, activates caspase 8, and leads to cell death by apoptosis. (3) Complex IIb-mediated apoptosis which is dependent on RIPK1 activity. When no RIPK1 ubiquitination can occur, complex IIb or the riptosome is formed. This results in the induction of apoptotic cell death through a pathway similar to that of complex IIa. (4) Complex IIc (Necrosome)-mediate necroptosis. When deubiquitinated RIPK1 is present and caspases are inactivated, necroptosis will occur as a rescue mechanism, since caspase 8-dependent apoptosis can not happen. The execution of necroptosis is dependent on the kinase activity of both RIPK1 and RIPK3, which activate downstream protein MLKL.

**Figure 5 ijms-24-00731-f005:**
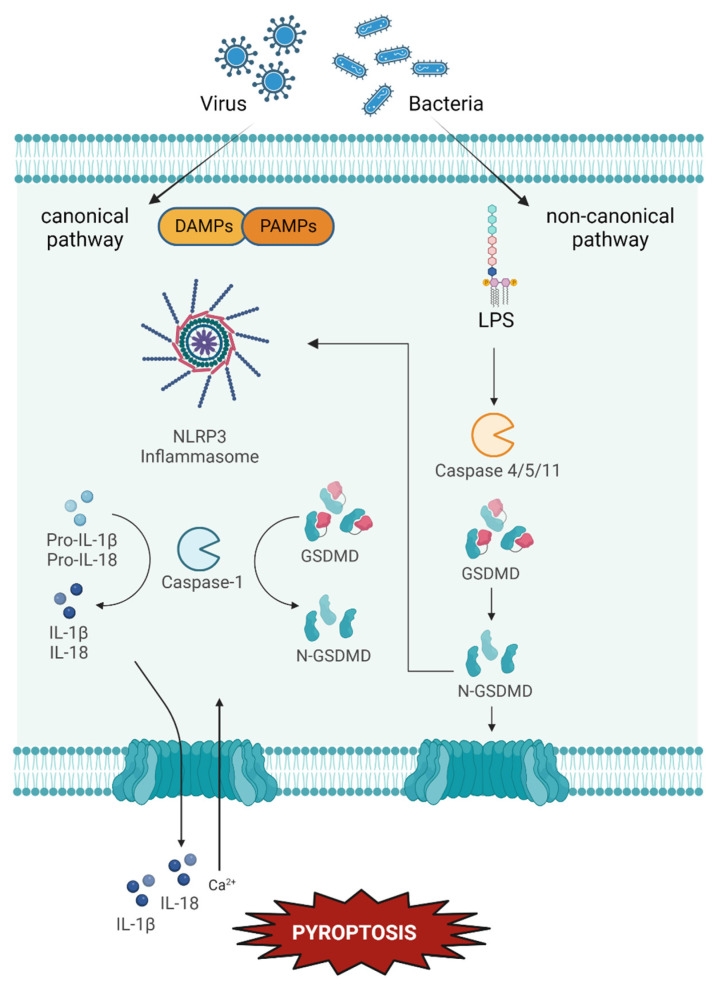
Overview of the pyroptosis pathway. After external stimulation mediated by pathogens, pyroptosis can occur via a canonical or a non-canonical pathway. In the canonical pyroptosis, following the external stimuli, the release of DAMPs or PAMPs can induce the formation of a multiprotein complex, inflammasome. The most studied is NLRP3. Inflammasome mediates the activation of CASP-1, which induces the maturation of IL-1β and IL-18 and the cleavage of GSDMD in the N-terminal fragment GSDMD-N responsible for the pore formation in the phospholipids bilayer. In the non-canonical pyroptosis, microbial LPS induces the oligomerisation of caspase 4/5 (human) or 11 (mouse), which mediates GSDMD cleavage in GSDMD-N and pore formation without enabling maturation of interleukins. At the same time, GSDMD activation via the non-canonical pathway can also promote the amplification of pyroptosis, stimulating the canonical pathway.

**Table 1 ijms-24-00731-t001:** Overview of the different DED biomarkers of the ocular surface and their possible connection with the different RCD.

DED Biomarkers on the Ocular Surface	Biomarkers [73]	Possible Correlation with RCD
**Inflammatory biomarkers** [74]	TNF-α	✓ Necroptosis
	MMP-9	✕ (Apoptosis)
	IL-1β	✓ Pyroptosis
	IL-6	✓ Necroptosis
	IL-17A	✕ (Apoptosis)
	IL-18	✓ Pyroptosis
**Tear film biomarkers** [75,76,77]	ROS	✓ Ferroptosis, necroptosis
	Hyperosmolarity	✓ Ferroptosis
	Lipid peroxidation	✓ Ferroptosis
	↓Lactoferrin	✕ n.d.*
	↓Lysozyme	✕ n.d.*
	Chemokyne/cytokines	✓ Necroptosis, pyroptosis

* n.d. = not defined. It is not associated with a specific cell death mechanism.

## Data Availability

No new data were created or analyzed in this study. Data sharing is not applicable to this article.

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
