# Peer review of "The Potential Role of Regulated Cell Death in Dry Eye Diseases and Ocular Surface Dysfunction"

_ijms, 2023, doi:10.3390/ijms24010731_

Round 1

Reviewer 1 Report

Manuscript Summary: 

In this article, the authors performed a literature review on different types of regulated cell death mechanisms such as ferroptosis, necroptosis and pyroptosis in dry eye disease and ocular surface dysfunction. The authors reviewed the fundamental reasons for dry eye disease (DED) including oxidative stress and inflammation and provided different mechanisms of regulated cell death and possible therapeutics for DED. The authors have presented a relevant article through systematic analysis of the literature; however, the following comments need to be addressed by the authors to enhance the review manuscript: 

Major comments: 

1.    References are missing in the statements written in the introduction such as lines 22-23 and lines no. 32-33.  Likewise, references are missing many of the statements throughout the text. It would be important to reference wherever scientific statements have been made throughout the manuscript.

2.    In line 37-39, the authors mentioned “However, recent studies hypothesized that menopause evaporative DED was actually due to the reduction of androgens levels rather than estrogens [6].” However, only one reference is quoted. 

4.    For lines 237-239, the authors mentioned “Recent studies reported the implication of ferroptosis in the occurrence and progress of many diseases such as tumors, neurological diseases, acute kidney injury, and ischemia/reperfusion”. But they have quoted only a single reference for this. 

5.    In line no. 250, the authors mentioned “Different studies underlined how the loss of cell functions is an important contribution to the alteration of the ocular barrier”. But the authors provided only one reference for this.  

6.    In line no. 251, The authors mentioned, “We hypothesize therefore that ferroptosis can be considered a key mechanism in the etiopathology of DED”. Please remove the reference for this if the statement is hypothesized by the authors.  

7.    Reference is missing for the statement written in lines 292-293  

8.    In line no. 304, provide reference 108. i.e., after “conjunctival cells from cytotoxicity (108)” and remove 108 from line no. 306.  

9.    In lines 340-341, the authors have written “Upregulation of RIPK1 was detected in ischemia-reperfusion injury (IRI), atherosclerosis, acute kidney injury, and neurodegenerative diseases”. But only one reference was provided for all these studies. Likewise, in 342-343, the authors mentioned “Recently, different studies revealed the possible implication of necroptosis in ocular conditions, with particular attention to corneal surface dysregulation”, but quoted only one reference. 

10.  Italicize the word in-vitro and in vivo, line no. 441, line no. 448. 

Minor comments 

  • Need to pay attention to minor formatting errors. 

Eg. Remove coma (,) before the reference in line no. 295,  

Denote Calcium ion as Ca2+ instead of Ca2+ in line no. 333 

Author Response

  1. References are missing in the statements written in the introduction such as lines 22-23 and lines no. 32-33.  Likewise, references are missing many of the statements throughout the text. It would be important to reference wherever scientific statements have been made throughout the manuscript.

The references are updated and completed based on the reviewer's comments.

  1. In line 37-39, the authors mentioned “However, recent studies hypothesized that menopause evaporative DED was actually due to the reduction of androgens levels rather than estrogens [6].” However, only one reference is quoted. 

Additional studies to support the statement are cited in the text.

  1. For lines 237-239, the authors mentioned “Recent studies reported the implication of ferroptosis in the occurrence and progress of many diseases such as tumors, neurological diseases, acute kidney injury, and ischemia/reperfusion”. But they have quoted only a single reference for this. 

The previous reference was a review that included all the mentioned pathologies where necroptosis was detected. However, following the suggestions of comment 3, additional studies are cited accordingly.

  1. In line no. 250, the authors mentioned “Different studies underlined how the loss of cell functions is an important contribution to the alteration of the ocular barrier”. But the authors provided only one reference for this. 

Additional studies to support the statement are cited in the text.

  1. In line no. 251, The authors mentioned, “We hypothesize therefore that ferroptosis can be considered a key mechanism in the etiopathology of DED”. Please remove the reference for this if the statement is hypothesized by the authors.  

Since the statement was hypothesised by the authors, the reference is removed.

  1. Reference is missing for the statement written in lines 292-293 

The corresponding reference is added.

  1. In line no. 304, provide reference 108. i.e., after “conjunctival cells from cytotoxicity (108)” and remove 108 from line no. 306.  

The references are adjusted as suggested by comment 7.

  1. In lines 340-341, the authors have written “Upregulation of RIPK1 was detected in ischemia-reperfusion injury (IRI), atherosclerosis, acute kidney injury, and neurodegenerative diseases”. But only one reference was provided for all these studies. Likewise, in 342-343, the authors mentioned “Recently, different studies revealed the possible implication of necroptosis in ocular conditions, with particular attention to corneal surface dysregulation”, but quoted only one reference. 

The corresponding references in support of the statements are added.

  1. Italicize the word in-vitro and in vivo, line no. 441, line no. 448. 

All the in vivo and in vitro words in the text are italicised.

Need to pay attention to minor formatting errors. 

    Eg. Remove coma (,) before the reference in line no. 295,  

    Denote Calcium ion as Ca2+ instead of Ca2+ in line no. 333

The comma was removed and the 2+ of Ca is superscripted.

Reviewer 2 Report

The work “The potential role of regulated cell death in dry eye diseases and ocular surface dysfunction” by Scarpellini et al. is an excellent review concerning regulated cell death (RCD) processes and its possible implications in ocular surface disease. The authors do a good job at presenting and defining the concept of regulated cell death and differentiating apoptosis from inflammatory cell death. Later they explain the RCD processes of ferroptosis, necroptosis, and pyroptosis and describe their possible associations with dry eye disease and ocular surface disorders. Moreover, they mention other ocular diseases where these have been evidenced. This is an important review (as the author suggest) because there is a need for novel therapeutic agents, which can derive from further understanding of the aforementioned mechanisms. It is overall well-written and should be considered for publication.

I have minor comments:
Line 101-120: On the section on current anti-inflammatory medications. The authors mention partial information about these:
- Cyclosporine 0.05 is denominated “Restasis”.
- There is no mention of cyclosporine 0.09 “Cequa”

-          Xiidra was approved for signs and symptoms of DED, unlike the previous 2 (approved for signs only)

-          There is another drug which is loteprednol etabonate ophthalmic suspension named “Eysuvis” (approved for inflammatory flares)

-          Could the authors hypothesize the role that these anti-inflammatory medications (steroidal vs non-steroidal) may have on the 3 regulated cell death mechanisms mentioned?

Line 134: hydroperoxides seems to correlate well with other… Please state if the correlation is positive or negative.

Line 266: should be written “induced”

Line 435/436: Pyroptosis seems to be an inflammatory process involved in DED. In dry eye, the patients had elevated GSDMD-N concentrations [156]. Please rephrase.

Line 443: The dry eye symptoms, such as tear loss or inflammatory response.
These are not dry eye symptoms, but rather signs.

Author Response

Line 101-120: On the section on current anti-inflammatory medications. The authors mention partial information about these:
- Cyclosporine 0.05 is denominated “Restasis”.
- There is no mention of cyclosporine 0.09 “Cequa”

-          Xiidra was approved for signs and symptoms of DED, unlike the previous 2 (approved for signs only)

-          There is another drug which is loteprednol etabonate ophthalmic suspension named “Eysuvis” (approved for inflammatory flares)

The list of the approved medication for DED are adjusted and updated based on the first comment.

-           the authors hypothesize the role that these anti-inflammatory medications (steroidal vs non-steroidal) may have on the 3 regulated cell death mechanisms mentioned?

Generally, an effective strategy to inhibit the different RCD, is the inhibition of upstream mediators rather than contrasting the inflammation itself.

Considering the specific target for each of the reported medications, it is possible to hypothesise an interference of Xiidra with necroptosis, due to the cytokines releases mediated by RIPK1 activation and prevention of the necroptosis mechanism itself which is commonly induced by TNF-alpha. Similarly, we can assume that the medication could also interfere with the release of the cytokines derived from pyroptosis activation. Evidence in the literature reported Cyclosporine A (CsA) as a necroptosis inducer.1 On the opposite, the study reported by Fakharnia F et al., reported cyclophilin D to promote necroptosis and the treatment with CsA seems to reduce the expression of necroptosis markers.2 It is important to mention that none of the models was ascribable to eye diseases. Restasis seems to interfere with cyclophilin D on the mitochondria membrane preventing its dysfunction. In a recent study, cyclophilin D-deficient cells died after apoptotic stimuli, but they resisted necrotic inducers and radical oxygen species overload. This evidence would bring to the assumption that Restasis, due to the specific mechanism of action, could prevent the inflammation mediated by the three RCD analysed in our review.3

For the ferroptosis mechanism itself, the use of antioxidants is still the best strategy. Probably, the use of both, steroidal and non-steroidal anti-inflammatory medications has some limitations. In our opinion, as general comments for the three described RCD, there is not sufficient literature to conclude whether their application might have a beneficiary therapeutic effect. However, we can speculate that whenever the release of the cytokine is directly tackled, the inflammation mediated by RCD might be prevented. This is in support of our review. Therefore, it is important to consider different RCD as potential novel mechanisms and/or as a mechanism target from one of the available dry eye medications. Since all this is highly hypothetical and is not supported by scientific literature, we decided to not change the manuscript.

  • Line 134: hydroperoxides seems to correlate well with other… Please state if the correlation is positive or negative.

The word “positively” is added in place of the word “well” as suggested.

  • Line 266: should be written “induced”

The word “induce” is corrected to “induced”.

  • Line 435/436: Pyroptosis seems to be an inflammatory process involved in DED. In dry eye, the patients had elevated GSDMD-N concentrations [156]. Please rephrase.

The sentence is rephrased: "The elevated level of GSDMD-N detected in dry eye patients, supports the hypothesis of pyroptosis implication in DED.”

  • Line 443: The dry eye symptoms, such as tear loss or inflammatory response.
    These are not dry eye symptoms, but rather signs.

The sentence is corrected based on the reviewer's suggestion.

References: 

  1. Ouyang Z, Zhu S, Jin J, et al. Necroptosis contributes to the cyclosporin A-induced cytotoxicity in NRK-52E cells. Pharmazie. 2012;67(8):725-732. doi:10.1691/ph.2012.1837
  2. Fakharnia F, Khodagholi F, Dargahi L, Ahmadiani A. Prevention of Cyclophilin D-Mediated mPTP Opening Using Cyclosporine-A Alleviates the Elevation of Necroptosis, Autophagy and Apoptosis-Related Markers Following Global Cerebral Ischemia-Reperfusion. J Mol Neurosci. 2017;61(1):52-60. doi:10.1007/s12031-016-0843-3
  3. Halestrap AP. Dual role for the ADP/ATP translocator? Nature. 2004;430(7003):984-984. doi:10.1038/nature02816

Reviewer 3 Report

I think it is a good review for the etiopathology and the treatment strategy of dry eye diseases.

Minor change:

The reference number mast be followed by citation order.

Therefore,

Line 258: reference number [106]à[94],

following this change, line 260 [96]à[95], line 263 [97]à[96], line 265 [94]à[97].

After reference number [107] to [129] are change to [106] to [128]

Line 343: reference number [95]à[129]

Please confirm the reference number in text and References.

Author Response

The reference number mast be followed by citation order.Therefore,

  • Line 258: reference number [106]à[94],
  • following this change, line 260 [96]à[95], line 263 [97]à[96], line 265 [94]à[97].
  • After reference number [107] to [129] are change to [106] to [128]
  • Line 343: reference number [95]à[129]

The reference numbering is adjusted based on the comments received.